# Rattlesnake Roundup: Point-of-Care Thrombelastographic Methods Define the Molecular Impacts on Coagulation of *Crotalus* Venom Toxins In Vitro and In Vivo

**DOI:** 10.3390/toxins17020087

**Published:** 2025-02-13

**Authors:** Vance G. Nielsen

**Affiliations:** Department of Anesthesiology, College of Medicine, University of Arizona, Tucson, AZ 85750, USA; vnielsen@anesth.arizona.edu

**Keywords:** rattlesnake venom, serine protease, metalloproteinase, fibrinogen concentration, platelet count, thrombelastography, rabbit model, point-of-care methods

## Abstract

A malalignment between rattlesnake-envenomed patients’ degree of compromised coagulation and the data generated by standard hematological determinations generated with blood samples anticoagulated with calcium (Ca) chelating agents is almost certain. Many rattlesnake venom toxins are Ca-independent toxins that likely continue to damage plasmatic and cellular components of coagulation in blood samples (anticoagulated with Ca chelation) during transportation from the emergency department to the clinical laboratory. The most straightforward approach to abrogate this patient–laboratory malalignment is to reduce “needle to activation time”—the time from blood collection to commencement of laboratory analysis—with utilization of point-of-care (POC) technology such as thrombelastography. The workflow and history of standard and POC approaches to hematological assessment is reviewed. Further, using a preclinical model of envenomation with four different rattlesnake venoms, the remarkably diverse damage to coagulation revealed with POC thrombelastography is presented. It is anticipated that future investigation and potential changes in clinical monitoring practices with POC methods of hematological assessment will improve the management of envenomed patients and assist in precision care.

## 1. Introduction

The field of rattlesnake envenomation can be a kaleidoscope of clinical, preclinical, and molecular bodies of information that can align or conflict with each other [1,2,3,4,5,6,7,8,9,10,11,12,13,14,15,16,17,18,19,20,21,22,23,24,25,26,27,28,29,30,31,32,33,34,35,36,37,38,39,40,41,42,43,44,45,46,47]. Rattlesnake venom is complex, containing metalloproteinases (SVMPs), serine proteases (SVSPs), thrombin-like enzymes (SVTLEs), phospholipaseA_2_ (SVPLA_2_), extracellular vesicles, and nonenzymatic compounds such as c-type lectins and disintegrins [1,2,3,20,21,22,23,24,25,27,28,29,30]. In addition to vast diversity in molecular target, these compounds within rattlesnake venom also have variability in biochemical requirements to be biologically active, such as being calcium (Ca)-independent [1,2,3,31,39] or Ca-dependent [1,2,3,34,35,38]. This diversity in venom proteome has significant implications for not just the clinical outcomes associated with rattlesnake envenomation but also markedly impacts the laboratory monitoring, antivenom administration, and experimental designs with in vitro and in vivo models. While the diagnosis and treatment of rattlesnake envenomation is based on clinical symptoms such as pain, edema and ecchymosis at the bite site, and the speed of spread of such tissue damage [47], the use of clinical hematological laboratory assessments (e.g., fibrinogen concentration, platelet count) are routinely made [4,5,6,7,8,9,10,11,12,13,14,15,16,17,18,19]. The severity of envenomation is often assessed by the degree of derangement of the laboratory values, with reports of no fibrinogen [12,16] detected or marked thrombocytopenia noted [5,6,7,15]. Persistent or recurrent derangements of coagulation are also commonly reported with standard hematological methods [5,18,19]. While these investigations provide in detail the clinical laboratories used to assess the severity of envenomation, to determine the degree of laboratory-based coagulopathy, and to measure the efficacy of antivenom administration [4,5,6,7,8,9,10,11,12,13,14,15,16,17,18,19], key details that are the underpinning of the biological relevance of the laboratory values generated are missing. There are marked degrees of uncertainty caused by the interplay of snake venom toxins, blood sample collection, and hematological methodology that disengage laboratory results from patient condition that have been completely unappreciated. One of the goals of this review is to raise awareness of this problem and introduce the concept of “needle to activation” as a new goal for both clinicians and laboratorians that treat and investigation envenomation, respectively. The reduction of time from blood sample collection to the activation of blood sample within a hematological parameter measuring device is the new challenge, with the use of point-of-care (POC) viscoelastic methods at the forefront of this endeavor. The other goal of this review is to demonstrate how this approach to coagulation assessment has been used in a preclinical, rabbit model of envenomation in the recent past utilizing venom from the North American rattlesnakes *Crotalus atrox* (*C. atrox*, Western diamondback rattlesnake), *C. scutulatus scutulatus* (*C. s. scutulatus*, Mojave rattlesnake, venom type B), and *C. adamanteus* (Eastern diamondback rattlesnake) [32,33,36,38]. Further, there will be the inclusion of previously unpublished data utilizing venom derived from *C. molossus molossus* (*C. m. molossus*, Northern blacktail rattlesnake). But, first, the reader should consider the problem posed by rattlesnake-envenomed patients without much clinical bleeding found to have laboratory values that are “bleeding to death”.

## 2. Swollen Patients, Bleeding Test Tubes

While there are always patients with clinical signs of envenomation that include edema and bleeding at the bite site, the assessment of the degree of compromise of coagulation is subject to uncertainty and variability, as will be subsequently presented. First, understanding the methodology involved in blood collection and standard hematological analyses needs to be considered. Specifically, whole-blood samples collected from patients are usually placed into a “blue top”, sodium (Na) citrate-containing tube for determination of activated partial thromboplastin time (aPTT), prothrombin time (PT), and fibrinogen concentration. Further, whole blood is also placed into a “purple top”, ethylenediaminetetraacetic acid-containing (EDTA) tube for assessment of platelet count. Both tubes are designed to prevent contact protein, factor XII-mediated thrombin generation via interactions of blood with the plastic surface of the tube by chelating Ca with Na citrate or EDTA, which prevents calcium-dependent prothrombin to thrombin conversion. This chelation allows sufficient time for transportation of the sample to the clinical laboratory. Additional centrifugation is required for Na citrate anticoagulated samples before analyses, whereas samples with EDTA may be analyzed upon arrival to the laboratory, as seen in Figure 1.

This methodology is simple and highly reliable when assessing the laboratory parameters displayed when evaluating patients with potential deficits in coagulation caused by most congenital and acquired hematological diseases. However, there is one tacit assumption made when considering these standard laboratory tests: chelation of Ca by Na citrate and EDTA quenches all hematological processes before the final analyses. And, for most clinically encountered scenarios, this approach is highly reliable when assessing patients. In fact, many in the readership may consider the use of Na citrate and EDTA the “gold standard” because of presumed stability of the sample. However, when it comes to venomous snakebite, there is a very serious problem with this method of hematological assessment that is subsequently presented.

Snake venom compounds, enzymatic and nonenzymatic, have activities and interactions with target molecules such as fibrinogen and platelet receptors that may be Ca-dependent or Ca-independent [1,22,23,24,25,31,34,35,36,37,38,39]. That means snake venom within the blood collected from a snake bite victim may or may not be destroying coagulation proteins in the Na citrate anticoagulated sample or aggregating (or preventing future aggregation) of platelets in the EDTA anticoagulated sample. Thus, depending on the variability in time for each process prior to analysis depicted in Figure 1, a coagulopathy may be created within the blood samples that clearly does not reflect what is occurring within the patient. This is not to say that the envenomed patient does not have compromised coagulation for clinical reasons, such as severe envenomation or delayed presentation (e.g., 16 h after envenomation [4]) prior to antivenom administration. Instead, the use of Ca chelation as a method of preservation of coagulation prior to transportation and processing/analyses remote from the area of patient treatment adds a degree of uncertainty to the alignment of the patient clinical situation with the laboratory analysis.

The most unambiguous demonstration of the degradation of coagulation in Na citrate anticoagulated human plasma by rattlesnake venom, in this instance obtained from *C. atrox*, is observed in [31], a publication nearly nine years old. On page 507, in Figure 1, it is observed that five minutes of exposure of Na citrated anticoagulated human plasma to 0.5 µg of venom resulted in significant degradation of coagulation as assessed by thrombelastography [31]. Further, and critically, exposure to the same concentration of venom for twenty minutes all but eliminated coagulation, as displayed on page 508, Figure 2 [31]. While these samples were exposed to a temperature of 37 °C [31], which is greater than room temperature, during transport from the bedside to the laboratory, it is highly likely that degradation of the sample by Ca-independent enzymes would occur after envenomation by this viper, as displayed in Figure 1 of this work. These data, using human plasma anticoagulated with Na citrate, serve as perhaps the strongest rationale to advocate for POC methods to assess coagulation at the bedside.

The same problem of in vitro compromise of coagulation by Ca-independent venom proteins presents itself in a reverse manner when antivenom is administered. Fibrinogen concentrations and platelet counts may rapidly recover after envenomed patients are treated, sometimes improving markedly with repeated administrations of antivenom [5,6,7,10,12,14,15,17,18,19]. Standard antivenom (e.g., intravenous, antibody-based) administration removes circulating venom, and thus there is less venom to compromise coagulation in the patient’s circulation and in their blood samples. Eventually, the effects of venom are abrogated when adequate antivenom is administered, and then—and only then—the in vitro standard hematological laboratory assessments are representative of the patient’s in vivo coagulation. This malalignment could lead to potential excessive antivenom administration or even unneeded transfusion of platelets/fresh frozen plasma/cryoprecipitate to treat a clinical coagulopathy that does not exist. So how does one eliminate or at least significantly diminish the divergence in the clinical state of the patient from compromised laboratory analyses following a venomous snake bite? A potential solution has been in plain sight for decades [48].

## 3. Reducing “Needle to Activation Time”

The simplest approach to aligning in vivo coagulopathy to laboratory hematological assessment is decreasing the time from blood collection to commencement of the laboratory test. This is one of the goals of POC evaluation, and a typical laboratory workflow diagram that illustrates the basic events and devices with a thrombelastographic system is displayed in Figure 2.

The first cases of POC thrombelastographic analysis with “native whole blood” (meaning without processing or activation) were reported in 1990, involving patients bitten by *Dispholidus typus* (Boomslang) in South Africa [48]. Native whole blood must be collected and placed into a thrombelastograph within 5 min or less to prevent activation of coagulation within the syringe used to collect the sample. Standard hematological assessments were also made, and, given that both patients presented with severe clinical bleeding 36 h after envenomation, both standard hematological tests and thrombelastographic data demonstrated severe loss of coagulation capacity by every measure made [48]. This is the first description of this approach, and it is of interest that *D. typus* venom activity is procoagulant, activating prothrombin [49,50], and is Ca-independent [50]. There is an earlier report of using thrombelastography to assess the effects of envenomation by a *Trimeresurus* species, but this report involved using blood that was anticoagulated with Na citrate that then had Ca added to commence data collection [51]. A year later, it was reported that similar *Trimeresurus* species had anticoagulant venom that was Ca-dependent [52], which was later observed utilizing similar *Trimeresurus* venoms with thrombelastography with human plasma [53]. Considered together, the first description of thrombelastographic assessment of envenomation was likely in alignment with the patient’s conditions, as the activity of the venom involved was Ca-dependent and therefore not interacting with the Na citrate anticoagulated blood sample prior to analysis [51]. In contrast, POC thrombelastographic analysis was quintessential to being aligned with the patient’s condition, given the Ca-independent activity of *D. typus* venom [48]. In summary, if the Ca-dependent status of the venom activity derived from a snake bite is unknown, then the approach most likely to result in data in alignment with the victim’s state of coagulation would be derived from POC thrombelastographic methods, such as has been used in numerous laboratory and preclinical investigations in recent years [31,32,33,34,35,36,37,38,39]. Put another way, a reduction in “needle to activation time” may be the key to success when assessing the impact of envenomation by North American rattlesnakes with markedly diverse venom characteristics [4,5,6,7,8,9,10,11,12,13,14,15,16,20,21,22,23,24,25,26,27,28,29,30,31,32,33,34,35,36,37,38,39].

## 4. A Rattle Is About the Only Feature North American Rattlesnakes Share

Rattlesnakes may have venom that is procoagulant, anticoagulant, platelet activating (causing aggregation), or platelet inactivating (preventing aggregation) in nature; this variability is markedly species-specific. For example, while toxicologists that treat bites from the Eastern diamondback rattlesnake found in the Southeastern United states would not have to worry about their patient having a clinical course like one bitten by a Western diamondback rattlesnake in Arizona, both venoms have Ca-independent activities confirmed with thrombelastography [31,37]. Worse yet, the hemotoxic venom activity of the Mojave rattlesnake is Ca-dependent [34,35,36], and the Western diamondback rattlesnake, with Ca-independent activity, occupies the same territory in Arizona. The fibrinogenolytic activity of both Western diamondback rattlesnake and Mojave rattlesnake venoms compromise plasmatic coagulation in vitro and in vivo, but only the venom of the Western diamondback rattlesnake compromises platelet function as determined by thrombelastography [31,32,33,34,36]. Whole-blood coagulation is not compromised by Mojave rattlesnake venom, while plasmatic coagulation is significantly decreased [36], demonstrating that, while fibrinolytic enzymes damage the ability of fibrinogen to be polymerized by thrombin, the ability of platelets to bind fibrin polymers with glycoprotein IIb/IIIa receptors are not. In contrast, the venom of the Eastern diamondback rattlesnake contains a serine protease with thrombin-like activity that polymerizes fibrinogen as a weak clot, depleting the fibrinogen concentration available for thrombin polymerization [37]. This results in a simultaneous decrease in whole-blood and platelet-inhibited blood coagulation kinetics in vivo, as demonstrated via thrombelastography in the rabbit [39]. The loss of fibrinogen decreases fibrin polymer formation, which, in turn, decreases binding of platelet glycoprotein IIb/IIIa receptors that results in a loss of whole-blood clot strength that is platelet-mediated with Eastern diamondback rattlesnake venom that does not contain any antiplatelet activity [14,28,39]. Thus, the thrombelastographic profile of envenomation of rabbits by the Western diamondback rattlesnake or Eastern diamondback rattlesnake will be nearly identical but for completely different biochemical reasons [32,33,39]. Figure 3 depicts images of these three rattlesnakes, representative venom activities to the right of each viper, and then observed changes in coagulation kinetics and corresponding thrombelastographic parameters.

In conclusion, diversity in venom toxins, especially in Ca-dependent/independent activities, in the species of venomous snake that share the same geographical area and threaten the same human population serves as a rationale for using POC thrombelastographic methodology to assess coagulopathy. Finally, to add to the data presented generated from these three rattlesnakes, de novo data obtained with *C. m. molossus* venom will be presented. This viper shares the same territory in Arizona as the Western diamondback rattlesnake and the Mojave rattlesnake.

## 5. *C. m. molossus*

*C. m. molossus* is a medium-sized, relatively docile rattlesnake that is purported to have a primarily hemotoxic venom that is “two-thirds” as potent at the western diamondback [40,41,42]; however, this author has not been able to confirm this ratio of potency between the two species with any scientific reference. Instead, in an investigation using lethal dose 50% (LD_50_) in mice as the outcome, it was prospectively noted that *C. atrox* venom had an intravenous LD_50_ value of 3.79 mg/kg, whereas *C. m. molossus* venom had an LD_50_ value of 4.42 mg/kg [54]. Further, these investigators noted other works had reported that *C. atrox* venom had an intravenous LD_50_ value of 3.74 mg/kg, whereas *C. m. molossus* venom had an LD_50_ value of 2.04 mg/kg [54]. Based on the variation of murine LD_50_ values, there did not seem to be that much difference between the two venoms. Further, and intriguingly, there have only been two case reports of envenomation by *C. m. molossus* [15,16], with the first documenting severe pain, swelling and ecchymosis of the bitten limb, and coagulopathy featuring thrombocytopenia and hypofibrinogenemia in two adult men (31 and 47 years old) that required significant antivenom administration and hospitalization for 7–10 days [15]. This report also featured an in vitro study of the venom, with the authors verifying fibrinogenolytic activity and the ability to cause platelet aggregation [15]. In sharp contrast, a 12-year-old female bitten by an adult *C. m. molossus* demonstrated no changes in coagulation, instead exhibiting lower limb pain and edema that easily resolved with antivenom administration, resulting in discharge from the hospital in two days [16]. The rattlesnakes inflicting the bites were adults [15,16]. Considered as a whole, these investigations involving either murine LD_50_ [54] or clinical and laboratory outcomes [15,16] potentially demonstrate variability in potency following envenomation by *C. m. molossus*, beyond just being two-thirds as potent as *C. atrox* [40,41,42].

Adding to the paucity of data available concerning *C. m. molossus* are the few and relatively older biochemical investigations of its venom [43,44,45,46]. Two early works identified metalloproteinases within the venom which were named M4 [43] and M5 [44] that were found to cleave the Aα and Bβ chains of fibrinogen and the α and β chains of fibrin. Later work isolated another fibrinogenolytic metalloproteinase, CMM4 [45], from the venom, which was also found to be one of the most potent hemorrhagic enzymes identified. Hemorrhagins destroy basement membranes, allowing bleeding into envenomed tissues. Of particular interest, an investigation that assessed the variation in fibrinolytic- and complement-activating properties between venoms obtained from variable sizes of *C. m. molossus* vipers found that smaller snakes had far less potential hemotoxicity compared to larger specimens [46]. It should also be noted that the only work addressing *C. m. molossus venom*-mediated effects on platelet aggregation was presented in the first case report [15], which did not provide any detailed biochemical characterization of the compounds responsible. The components of venom that affect platelet activity include nonenzymatic proteins such as disintegrins that prevent aggregation, as found in *C. atrox* venom [22], or c-type lectins that cause aggregation [3]. In summary, while some preliminary investigation has been performed addressing the mechanisms responsible for the hemotoxicity of *C. m. molossus* venom, critical gaps in knowledge remain.

Given this background, the goals of the present study were as follows: (1) to utilize a validated rabbit model of subcutaneous envenomation [33,36,39] to characterize the effects of *C. m. molossus* venom on whole-blood coagulation, without and with platelet inhibition, and (2) to test the hypothesis that site-directed ruthenium-based antivenom administration could significantly attenuate the effects of *C. m. molossus* venom in rabbits. Ruthenium-based antivenom is a mixture of inorganic ruthenium-containing compounds in a mixture of organic and inorganic solvents that acts by binding key amino acid residues in several venom toxins—the readership may obtain greater detail from [33,36,39]. The source of venom, hemotoxic constituents, and likely affected thrombelastographic parameters are displayed in Figure 4.

## 6. Results

### 6.1. General Observations

Heart rate (168–242 beats/min average range) and arterial oxygen saturation (92–100% average range) values observed for all the animals were within the normal ranges reported previously with this model with other venoms [33,36,39]. The area of skin with venom injection had an area of ecchymosis (approximately 4–6 cm by 2–3 cm) that was oval in appearance. This ecchymosis developed over the hour of observation and did not appear to bother the rabbit, likely secondary to the local anesthetic administered prior to venom injection. The one rabbit that was administered ruthenium-based antivenom had a very pale ecchymotic area with a thin, much darker border (1–2 mm).

### 6.2. Dose–Response of C. m. molossus Envenomation on Whole-Blood Coagulation

Administration of 4, 12, 18, or 24 mg/kg of *C. m. molossus* venom subcutaneously did not change coagulation in this model (n = 1 per dose). However, 40 mg/kg seemed to affect coagulation; thus, it was found that envenomation to this degree (n = 6) resulted in significantly increased whole-blood TMRTG values (58%) and decreased whole-blood MRTG values (35%) and TTG values (17%) one hour after envenomation, as seen in Figure 5. Based on the similar decrease in whole-blood coagulation, *C. m. molossus* venom is ten-fold less potent than *C. atrox* venom [33]. Platelet-inhibited samples demonstrated a significant increase in TMRTG values (31%) and a significant decrease in MRTG values (33%); however, there was no significant change in TTG values, as noted in Figure 5. Critically, the %TTG_Platelet_ value decreased significantly from 86.1 ± 1.6% to 84.1 ± 2.1%.

Unfortunately, given the ten-fold greater amount of venom required to conduct these experiments than was anticipated by the author, there was only enough venom remaining for one animal experiment with antivenom administration and for in vitro investigation, subsequently described. The author viewed the in vivo experiment with ambivalence, as the concentration and quantity of the ruthenium-based antivenom permitted to be administered to rabbits of this weight class would likely be inadequate to neutralize the largest dose of venom ever administered in this model. Nevertheless, in the hope of obtaining insight, the experiment was conducted, with the results displayed in Table 1.

Administration of the ruthenium-based antivenom resulted in whole-blood samples that had TMRTG values that increased 29%, MRTG values that did not change, TTG values that increased 0.9%, and %TTG_Platelet_ that decreased by 0.2%. These changes were far less than that observed in whole blood obtained from rabbits not administered antivenom in Figure 5. The platelet-inhibited samples of this animal demonstrated an increase in TMRTG values (25%), a decrease in MRTG values (34%), and an increase in TTG (2%). These changes were not very different from those observed in animals not administered antivenom in Figure 5. Thus, under these circumstances, it appeared that ruthenium-based antivenom attenuated the effects of venom on platelet strength but did not modify the effects on plasmatic coagulation.

### 6.3. Effects on C. m. molossus Venom on Human Plasmatic Coagulation and Attenuation of Venom Effects by RuCl_3_

The rationale for this series of experiments was to determine if the relatively small potency of this venom on rabbit coagulation was species-specific and to determine if a component of the antivenom, RuCl_3_, would inhibit the fibrinogenolytic effects in a manner like that recently published with *C. atrox* venom [31], including determining if *C. m. molossus* venom activity was Ca-independent. It should be noted that the materials, including plasma, used in those results [31] were the same as in the present study. Preliminary experiments demonstrated that a *C. m. molossus* venom concentration of 24 µg/mL was required to diminish human plasmatic coagulation to the same extent as 3 µg/mL *C. atrox* venom [31]. The results of these experiments are seen in Figure 6.

The venom of *C. m. molossus* degraded Na citrate anticoagulated human plasma prior to the addition of Ca to activate coagulation, demonstrating Ca-independent activity. The venom significantly increased TMRTG values (67%), decreased MRTG values (88%), and decreased TTG values (60%) compared to the control values. In sharp contrast, venom exposed to RuCl_3_ generated thrombelastographic parameter values significantly different from the other two conditions, with greater TMRTG values (28%), smaller MRTG values (55%), and smaller TTG values (19%) compared to the control conditions. Thus, this venom was eight-fold less potent than *C. atrox* in human plasma in vitro [33]. Lastly, RuCl_3_ significantly inhibited the effects of *C. m. molossus* venom in vitro, verifying that the enzymes involved in the degradation of plasmatic function in rabbit plasma (Figure 5, Table 1) were likely not inhibited in the one rabbit administered antivenom secondary to inadequate concentrations of ruthenium-based antivenom.

## 7. Discussion

The de novo data derived from the characterization of *C. m. molossus* venom contribute to the impression that rattlesnake venom toxins are remarkably diverse, which, in turn, serves as a rationale for the use of POC thrombelastographic or other POC rapid hematological assessments to determine the degree of coagulation compromise of a snake bite victim. In the case of *C. m. molossus* venom, the thrombelastographic profile of loss of clot strength being primarily caused by decreased platelet-mediated clot strength and derangements in plasmatic coagulation kinetics without involving loss of plasma-mediated strength has been observed in rabbits administered intravenous *C. atrox* venom [33]. The loss of platelet-mediated clot strength caused by *C. atrox* venom may be secondary to loss of responsivity secondary to a disintegrin [22], aggregation by a c-type lectin [25], and, likely, both. In contrast, platelet aggregation by *C. m. molossus* venom as originally reported argues strongly for a primarily c-type lectin-mediated effect in human platelets [15] by this venom and in the inhibition of the rabbit whole-blood sample platelets of the present study. In contrast to the explainable effects of *C. m. molossus* and *C. atrox* venoms on platelets, the impact of the fibrinogen-modifying enzymes on the plasmatic coagulation of *C. m. molossus*, *C. atrox* [32], and *C. s. scutulatus* [36] are qualitatively different. *C. m. molossus* and *C. atrox* [32] prolong TMRTG values and MRTG values without affecting TTG, whereas *C. s. scutulatus* venom only significantly decreases MRTG and TTG values without changing TMRTG values [36]. Given that the progressive loss of fibrinogen from a plasma sample first decreases MRTG values and TTG values before TMRTG values increase in human plasma [55], the modifications of fibrinogen by the metalloproteinases contained in *C. m. molossus* and *C. atrox* venoms change, but do not perhaps always eliminate, fibrinogen as a substrate for thrombin. Unlike the static system of in vitro exposure of plasma to venom, such as in this work (Figure 5), in the living circulation of a rabbit, these enzymes are only briefly exposed to some but not all fibrinogen prior to being redistributed. Thus, a mixture of normal fibrinogen, partially catalyzed fibrinogen, completely catalyzed fibrinogen, and various fibrinogen split products are present in vivo. In contrast to *C. m. molossus* and *C. atrox*, *C. s. scutulatus* venom diminishes MRTG and TTG values in plasma but does not affect whole-blood coagulation (with platelet activity) [36], which in part is likely secondary to a modification of fibrinogen that does not meaningfully affect fibrinogen–platelet interactions. This concept of intact fibrinogen–platelet interactions occurring despite partial catalysis of fibrinogen by *C. s. scutulatus* venom is strengthened by the observation that the fibrinogen eliminating, not modifying, serine protease of *C. adamanteus* venom, a thrombin-like enzyme, has been observed to not only decrease plasmatic coagulation kinetics but also diminish whole-blood coagulation kinetics without any platelet inhibiting compounds present [39]. When these data are considered, knowledge of the toxins of these diverse *Crotalus* venoms coupled with the coagulation kinetic profiles of the venom effects on POC thrombelastographic parameter values allows greater mechanistic insight into the effects of such venoms on critical fibrinogen modifications/elimination and fibrinogen–platelet interactions.

There are limitations to the investigation of *C. m. molossus*, the most obvious being the inability to complete a full group of envenomed animals with antivenom treatment. Using all the venom in the acquired lot would have required obtaining far greater quantities of venom and performing at least twelve more experiments, which would have been a waste of animal life. Further, while there was not an apparent risk to the rabbits by the administration of such large quantities of venom, the administration of greater quantities of ruthenium-based antivenom would have required institutional approval and could have been potentially toxic, as recently reviewed in the earliest description of the model [38]. The paradigm to test envenomation and antivenom therapy was meant to simulate a bite on a rabbit-sized scale, but the bite was simulated with a 25 G needle, not large fangs, in a trauma-free manner. As recently reviewed [36], venoms can enter the central circulation a variety of ways, with lymphatic entry being predominant. However, if a venom proteome does not have the best combination of enzymes to gain lymphatic access, perhaps it is the intravenous absorption following the damage from a fang that facilitates systemic adsorption. The circumstances that the envenomed rabbits developed regional ecchymosis and required a tremendous amount of venom (40 mg/kg) to modestly affect coagulation may be evidence of the venom not being able to access the lymphatic system very well secondary to biochemical (lack of enzymes) or physical (venom trapped in the ecchymoses of tissues) impediments. The reality is that, if one was to treat an envenomation of an adult human being by this viper, the quantity of venom could never approach 40 mg/kg, and far more site-directed ruthenium antivenom could be safely administered to a human that may easily be 30–50-fold larger than a 2–3 kg rabbit. Another issue to consider is that not all venoms are meant to cause the death of their prey animals by severe, systemic coagulopathy. Instead, severe bleeding into the bite site and edema may cause shock and immobilization of prey. When considered as a whole, perhaps not all venoms are amenable to analysis with this rabbit model of minimal physical trespass, as it was designed to assess biochemical, not physical, contributions of envenomation to coagulopathy.

The goals of this review with inclusion of de novo data were to inform the readership about the malalignment of envenomed patient coagulation status with standard laboratory hemostasis tests and to use the example of rattlesnake envenomation of rabbits to illustrate the solution to this malalignment. In an environment containing vipers with diverse venom proteomes, there is uncertainty, especially regarding the Ca-dependence of key hemotoxins, and a reduction of “needle to activation time” is the most straightforward approach to eliminating this patient–laboratory value malalignment. However, it is unreasonable to advocate that all emergency departments should invest in the POC thrombelastographic system featured in this article (Figure 2), especially given the frequency of envenomation at even the busiest hospital. Fortunately, there are several small, modular, desktop devices that can generate critical data derived from an envenomed patient that use either fresh blood samples or Na citrate anticoagulated blood (analyzed just a few minutes after collection), utilizing just a drop of blood, that are relatively inexpensive and require less frequent calibration or maintenance. A few such devices are displayed for the reader’s convenience in Table 2.

While this review has focused on Ca-independent protein-mediated degradation of coagulation involving rattlesnake venom, this potential problem for coagulation assessment involves many species of snake across the globe. The venom of vipers from the genera *Agkistrodon* [56,57,58], *Bothrops* [59,60,61,62,63,64], *Echis* [65,66,67,68], *Lachesis* [69,70], *Micropechis* [71], *Naja* [72,73,74,75,76,77,78,79,80,81], *Oxyuranus* [82,83], and *Pseudonaja* [84,85,86] contains enzymatic and nonenzymatic proteins that can adversely affect plasmatic coagulation and platelet aggregation via Ca-independent mechanisms. There are many more examples, but the key issue is that assessment of coagulation with Na citrate or EDTA anticoagulated methods may have time-dependent degradation while awaiting assay worldwide.

In conclusion, it is anticipated that a reduction of “needle to activation time” with POC methodologies will provide new insights into the degree of damage done to envenomed patients’ coagulation. As can be observed with the POC thrombelastographic analyses of data derived from rabbits envenomed with four very different rattlesnake venom proteomes, insight into the nature of the toxin involved (e.g., platelet inactivation, fibrinogen-modifying) and the degree of coagulation compromise can be assessed without regard to the Ca-dependency of any specific toxin. Further, in addition to the findings of time-dependent, Ca-independent degradation of Na citrate anticoagulated human plasma sample coagulation by rattlesnake venom [31], a similar loss of coagulation was observed in Na citrate anticoagulated plasma exposed to cobra venom [81]. The thrombelastographic “finger print” of envenomation will depend on the effects of species-specific toxins, with different combinations of toxin potentially having the same “finger print.” Thus, as with all laboratory assessments, the clinical correlation with likely species responsible for envenomation must always be considered. The degree of inactivation of venom toxins by antivenoms, systemic (e.g., standard, antibody-based) or site-directed (e.g., ruthenium-based), may also be more accurately assessed, potentially obviating the need for the redosing of antivenom for concerns of coagulopathy. It is anticipated that future investigation and potential changes in clinical monitoring practices will improve the management of envenomed patients and assist in precision care.

## 8. Materials and Methods

### 8.1. Chemicals and Venoms

Lyophilized venom derived from an adult *C. m. molossus* was provided by the National Natural Toxins Research Center (NNTRC) located at Texas A&M University-Kingsville, Kingsville, TX, USA. The author purchased twice as much venom as previously needed based on recent investigations with rattlesnake venoms [33,36,39]. The National Institutes of Health fund the NNTRC out of the Office of Research Infrastructure Programs. Venom was dissolved into calcium-free phosphate buffered saline (PBS, Millipore Sigma, Saint Louis, MO, USA) to a final 60 mg/mL concentration, aliquoted, and maintained at −80 °C. This made the venom twice as concentrated as in previous investigations [35,38,41] as a precaution to make sure that the volume of venom injections would not become problematic with the rabbit model utilized [33,36,39] if the venom was markedly less potent than that of *C. atrox*. Pooled normal human plasma that was Na citrate anticoagulated and maintained at −80 °C was obtained from George King Bio-Medical (Overland Park, KS, USA). Dimethyl sulfoxide (DMSO), tricarbonyldichlororuthenium (II) dimer (CORM-2), and RuCl_3_ were obtained from Millipore Sigma (Saint Louis, MO, USA). Tissue factor for activating coagulation was obtained in the form of Pacific Hemostasis™ (Prothrombin Time Reagent, Thermo Fisher Scientific, Pittsburgh, PA, USA). CaCl_2_ (200 mM) was obtained from Haemonetics Inc. (Braintree, MA, USA).

### 8.2. Rabbit Model

Male New Zealand White rabbits (2–3 kg) were procured from Charles River Laboratories (San Diego, CA, USA) and housed within our animal facility and allowed food and water ad libitum for at least 1 week prior to experimentation. The reader is referred to the recent work for details concerning sedation, monitoring, envenomation, and antivenom administration [33]. The initial dose of *C. m. molossus* venom was based on the value of *C. atrox* venom previously used successfully in this model to cause coagulopathy [33], which was 4 mg/kg. The dose of venom was to be increased or decreased until a consistent pattern of coagulopathy was observed, which was then used for all subsequent experimentation. Blood samples were collected prior to venom injection and every hour for three hours in preliminary experiments, but this period was reduced to only one hour after venom injections after the dose for investigation was selected. If the animal was to be administered antivenom, then this occurred 5 min after injection of the venom, again administered through a 25G needle. After the last blood sample and vital sign assessments were obtained, the rabbits were euthanized with intravenous administration of 1 mL of pentobarbital/phenytoin (390/50 mg/mL) (Virbac, Westlake, TX, USA).

### 8.3. Coagulation Monitoring

#### 8.3.1. Rabbit Blood Assessments

Thrombelastographic analyses were performed as recently described in [36,39]. Blood samples (1 mL) were collected and immediately placed into a thrombelastographic cup for analysis with a computer-controlled thrombelastograph^®^ haemostasis system (Model 5000; Haemonetics Inc., Braintree, MA, USA) at 39 °C. The mixture used in the series of experiments was composed of 340 µL of whole blood, 10 µL of tissue factor (0.1% final concentration of Pacific Hemostasis™ Prothrombin Time Reagent, Thermo Fisher Scientific, Pittsburgh, PA, USA), and 10 µL of DMSO for samples without platelet inhibition or 10 µL of cytochalasin D (5 µM final concentration, dissolved in DMSO) in platelet-inhibited samples. After mixing the samples, the following parameters were determined: TMRTG (minutes), MRTG (dynes/cm^2^/second), and TTG (dynes/cm^2^). The %TTG_Plasma_ was calculated by dividing the TTG value of a platelet-inhibited sample by the corresponding TTG value of the whole-blood sample without platelet inhibition as previously described [38,41]. The value of %TTG_Platelet_ was calculated as 100% TTG_Plasma_. Data were collected for 30 min.

#### 8.3.2. Human Plasma Experiments

Plasma mixtures were placed in a disposable cup in a computer-controlled thrombelastograph^®^ haemostasis system, as described previously, at 37 °C [31]. The mixture used in the series of experiments was composed of 320 µL of plasma, 10 µL of tissue factor, 6.4 µL of PBS, 3.6 µL of venom solution, and, lastly, 20 µL of CaCl_2_. Plasma, tissue factor, and PBS were placed in the disposable cup, with venom solution added for 1 min prior to final addition of calcium chloride. Venom solutions were composed of 600–2400 µg/mL venom in PBS that had no addition or had RuCl_3_ placed as a 1% addition for a final concentration of 100 µM RuCl_3_. After the venom solutions were incubated at room temperature for 5 min, the aforementioned addition to the plasma mixture of the venom mixture was performed and the sample was mixed. Thus, the final concentration of venom in plasma was 6–24 µg/mL, which was expected to cause fibrinogenolysis like that of *C. atrox* [31]. After preliminary work, the three experimental conditions were a control condition, a venom condition, and a venom condition exposed to RuCl_3_ condition. This brief exposure was designed to result in a reproducible compromise of coagulation that could be quickly assessed by the rapid onset of tissue factor-initiated clot formation. After CaCl_2_ and final mixing of the samples, the previously described thrombelastographic parameters were determined. The data were collected for 15 min.

### 8.4. Statistical Analyses

The data are presented as mean ± SD. All experimental groups were represented by n = 6 different individuals, as this provides a statistical power > 0.8 with *p* < 0.05 using this methodology to assess differences in thrombelastographic parameters within and between groups [33,36,39]. The in vitro experiments had n = 4 per condition and had post hoc verification of a statistical power > 0.8 with *p* < 0.05. A commercially available statistical program was used for paired, two-tailed student’s *t*-tests or one-way analyses of variance (ANOVA) as appropriate to the dataset, with ANOVA followed by Holm–Sidak post hoc analyses (SigmaStat 3.1; Systat Software, Inc., San Jose, CA, USA). The graphics were generated with commercially available programs (Origen 2024, OrigenLab Corporation, Northampton, MA, USA, and CorelDRAW 2024, Alludo, Ottawa, ON, Canada). *p* < 0.05 was considered significant.

## Figures and Tables

**Figure 1 toxins-17-00087-f001:**
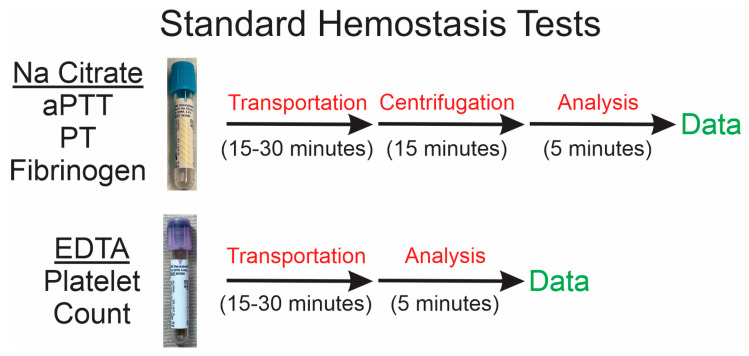
Timeline of events following blood collection prior to final data generation.

**Figure 2 toxins-17-00087-f002:**
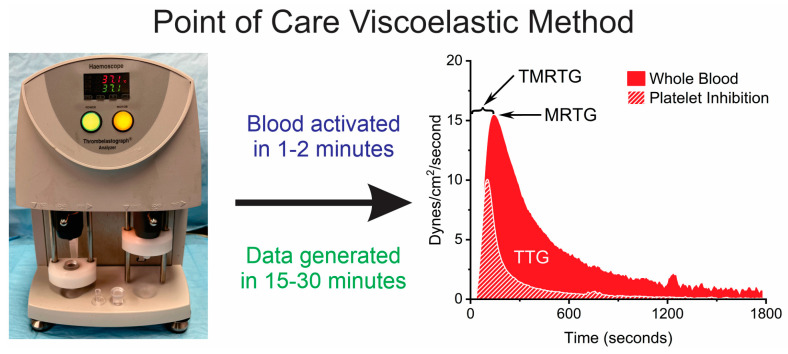
Thrombelastographic system using POC methodology and data generated. Blood is placed into cups with activators (e.g., tissue factor) ± platelet inhibitors within a few minutes after collection from a patient or preclinical model. Activation of coagulation occurs immediately, with data collected over 15–30 min. This radically reduces the “needle to activation time” of the analyses.

**Figure 3 toxins-17-00087-f003:**
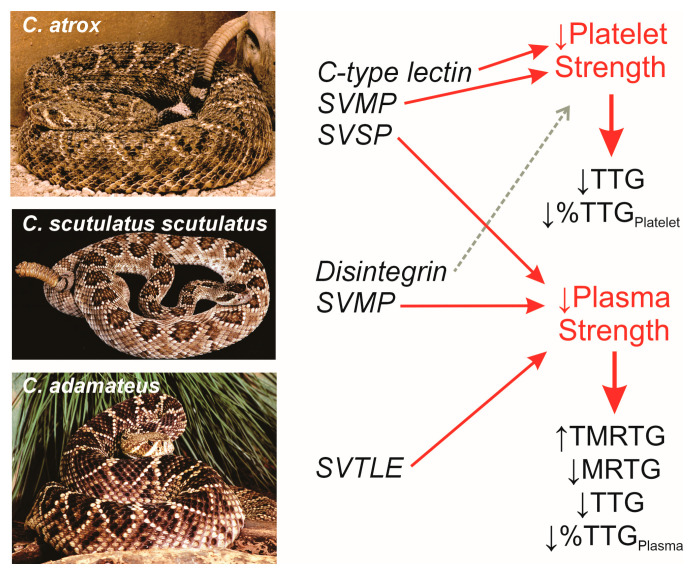
North American rattlesnakes with some venom toxins and observed thrombelastographic outcomes. Red arrows indicate the consequent feature of coagulation affected by the enzyme or compound indicated or describes the thrombelastographic parameter affected. The dashed gray arrow indicates an interaction found in vitro [27] but not in vivo [36]. TMRTG—time to the maximum rate of thrombus generation; MRTG (dynes/cm^2^/second)—the maximum rate of thrombus generation, the maximum velocity of clot growth observed; TTG (dynes/cm^2^)—total thrombus generation, the final viscoelastic resistance observed after clot formation. %TTG_Plasma_ is calculated by dividing the TTG value of a platelet-inhibited sample by the corresponding TTG value of the whole-blood sample without platelet inhibition; %TTG_Platelet_ is calculated as 100%–%TTG_Plasma_.

**Figure 4 toxins-17-00087-f004:**
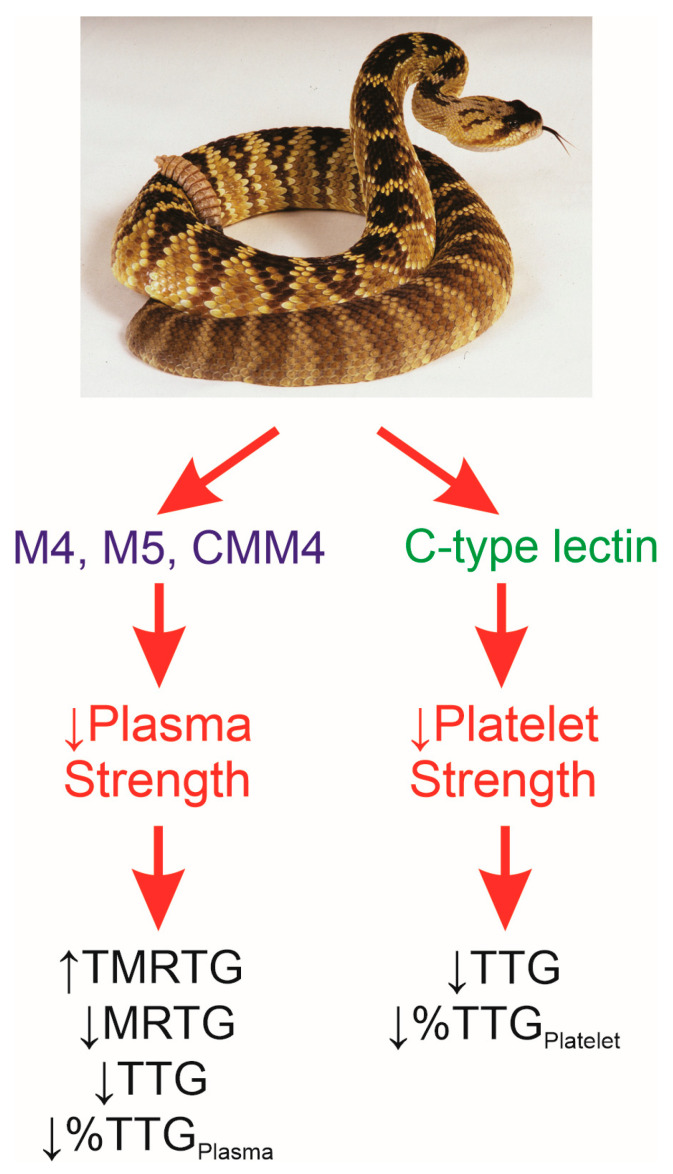
Venom source, components, and anticipated effects on thrombelastographic parameters. *C. m. molossus* is depicted at the top of the illustration; the enzymatic (green text) and nonenzymatic compounds (blue text) responsible for fibrinolysis and platelet aggregation found in the venom are subsequently displayed, and the expected increases or decreases in thrombelastographic parameters are depicted.

**Figure 5 toxins-17-00087-f005:**
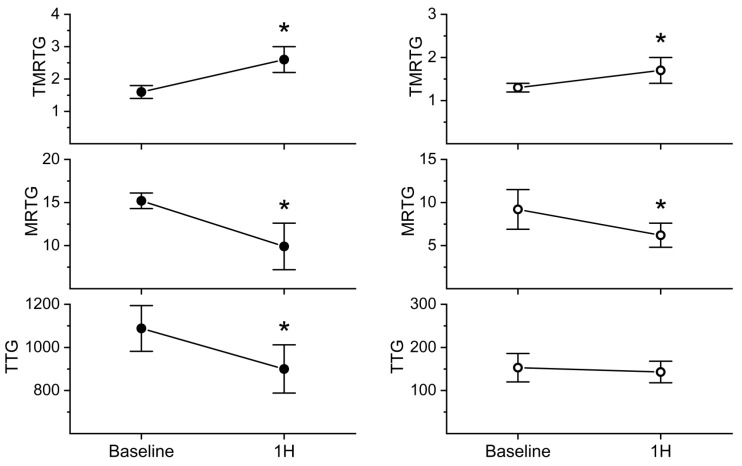
Effects of *C. m. molossus* venom injection without or with platelet inactivation on whole blood. The left panels represent data generated from whole blood obtained from animals injected with venom (closed black circles). The right panels represent data obtained from whole blood from the same rabbits after platelet inhibition of the samples. Data are presented as mean ± SD. TMRTG = time to maximum rate of thrombus generation (min); MRTG = maximum rate of thrombus generation (dynes/cm^2^/second); TTG = total thrombus generation (dynes/cm^2^). Time points are baseline before venom injection and one hour thereafter. * *p* < 0.01 vs. baseline.

**Figure 6 toxins-17-00087-f006:**
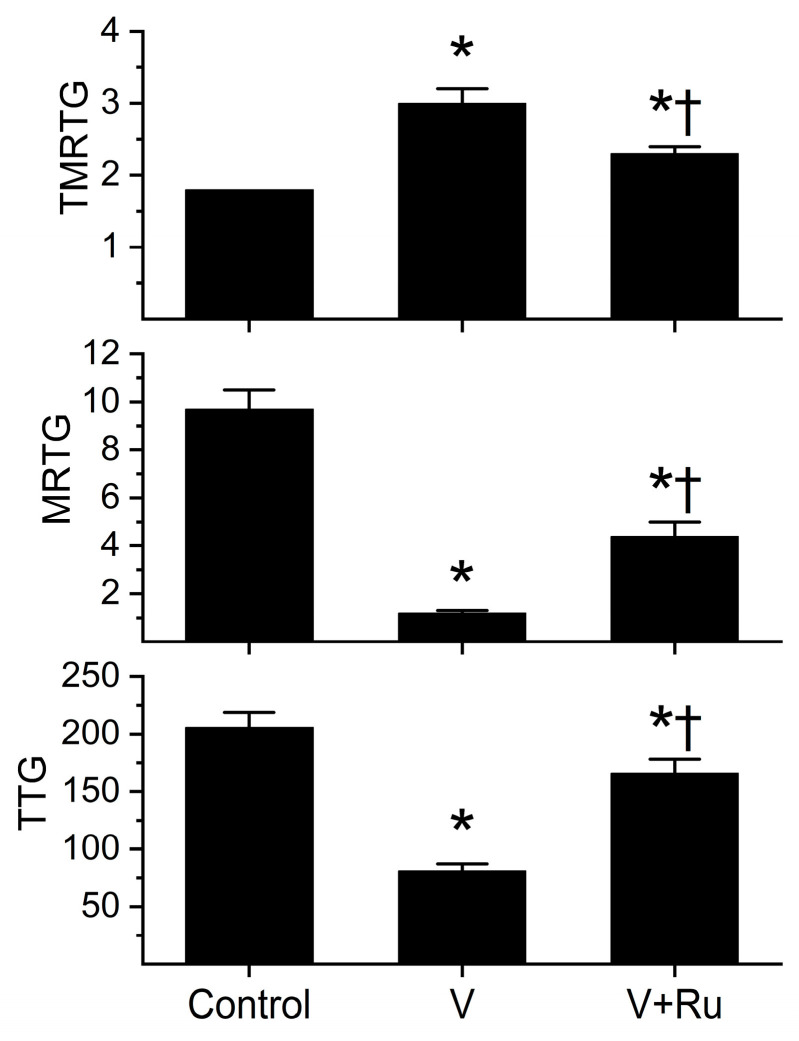
Effects of *C. m. molossus* venom on human plasmatic coagulation without or with inhibition by RuCl_3_. Control = plasma without additives; V = plasma with 24 µg/mL of venom; V + Ru = plasma with venom exposed to 100 µM RuCl_3_ prior to addition of the venom into the plasma. Data are presented as mean ± SD. * *p* < 0.001 vs. Control, † *p* < 0.001 vs. V.

**Table 1 toxins-17-00087-t001:** Effect of ruthenium-based antivenom on a rabbit envenomed with *C. m. molossus* venom.

Sample Type	Baseline	1 h
Whole Blood		
TMRTG	1.4	1.8
MRTG	13.7	13.7
TTG	978	987
%TTG_Platelet_	87.3	87.1
Platelet-Inhibited		
TMRTG	1.2	1.5
MRTG	7.3	4.8
TTG	124	127

**Table 2 toxins-17-00087-t002:** POC devices, data generated, and their manufacturers.

POC Device	Data	Manufacturer
PC100	Platelet Count	2M Engineering
qLabs FIB	Fibrinogen	Stago
Hemochron^®^ Signature Elite	PT, aPTT	Werfen
ROTEM sigma	Viscoelastic	Werfen
TEG^®^ 6s	Viscoelastic	Haemonetics Inc.

## Data Availability

The original contributions presented in this study are included in this article. Further inquiries can be directed to the corresponding author.

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
