# Peer review of "Rattlesnake Roundup: Point-of-Care Thrombelastographic Methods Define the Molecular Impacts on Coagulation of Crotalus Venom Toxins In Vitro and In Vivo"

_toxins, 2025, doi:10.3390/toxins17020087_

Round 1
Reviewer 1 Report
Comments and Suggestions for Authors
This paper reviews the role of Point-of-Care Thrombelastographic methods in monitoring coagulation index changes in patients with Crotalus envenomation and their correlation with the main toxic components, making the Point-of-Care detection of coagulation indicators in patients affected by snake bites instructive. However, I believe the paper has the following questions:
1. Could the authors explain or discuss the principles of the parameters TMRTG, MRTG, and TTG in Thrombelastographic methods, as well as the relationship between these parameters and the changes in coagulation factors in the blood? Many components of snake venom target specific proteins, such as thrombin-like enzymes that act on fibrinogen. Many studies have found that thrombin-like enzymes can reduce the fibrin content in the blood. Similarly, other enzymes also have this function. Besides the toxins that directly affect fibrinogen, there are many others that can influence other blood factors. Perhaps the parameters measured by these instruments are a combined result of the effects of various components in the snake venoms.
2. In Figure 5, the difference in TTG data between inhibited and non-inhibited platelets is highly significant. Can the authors analyze this from the perspective of the instrument characteristics and the toxin components? Are these indicators related to other tests such as APTT, PT, and platelet count?
3. While the data in Table 1 is important, it is based on a single animal's results. To maintain the rigor of the paper, it is suggested to remove this from results and briefly describe it in the discussion section with the data placed in supplementary materials.
4. The paper involves animal experiments. Has it received the necessary ethical approval? If so, please specify the approval situation or describe the ethical guidelines followed in the study.
Author Response
“This paper reviews the role of Point-of-Care Thrombelastographic methods in monitoring coagulation index changes in patients with Crotalus envenomation and their correlation with the main toxic components, making the Point-of-Care detection of coagulation indicators in patients affected by snake bites instructive. However, I believe the paper has the following questions:”
- “Could the authors explain or discuss the principles of the parameters TMRTG, MRTG, and TTG in Thrombelastographic methods, as well as the relationship between these parameters and the changes in coagulation factors in the blood?”
I appreciate the reviewer’s comment, but a comprehensive review of the first-derivative, parametric parameters of thrombelastography is outside the scope of this manuscript. My last reference in the revised work [57] was the beginning of several of my works over the next decade that defined the effects of procoagulant, anticoagulant, fibrinolytic, and antifibrinolytic proteins on thrombelastographic parameters. Thus, to do what the reviewer suggests would encompass another review article about viscoelastic methods such as thrombelastography and how parameters change with coagulopathy.
“Many components of snake venom target specific proteins, such as thrombin-like enzymes that act on fibrinogen. Many studies have found that thrombin-like enzymes can reduce the fibrin content in the blood. Similarly, other enzymes also have this function. Besides the toxins that directly affect fibrinogen, there are many others that can influence other blood factors. Perhaps the parameters measured by these instruments are a combined result of the effects of various components in the snake venoms.”
I appreciate the reviewer’s comments, but rattlesnake venom does not affect the serine proteases or other transamidases found in the coagulation pathways. The toxins of rattlesnake venom target fibrinogen and cause fibrinogenolysis or polymerize fibrinogen directly as is the case with TLE as in Eastern diamondback rattlesnake venom. The Clauss method measures fibrinogen levels generated in a diluted plasma sample in the presence of excess thrombin. So, if fibrinogen has undergone partial digestion by a fibrinogenolytic toxin, it will seem like the fibrinogen concentration is lower than it really is. However, TLE effectively remove fibrinogen from the circulation via polymerization, so the Clauss method will accurately report the actual fibrinogen concentration. With regard to the thrombelastograph, whatever enzyme/toxin is present with the greatest activity and highest rate of reaction will be revealed as the causative agent in both laboratory and clinical coagulopathy.
- “In Figure 5, the difference in TTG data between inhibited and non-inhibited platelets is highly significant. Can the authors analyze this from the perspective of the instrument characteristics and the toxin components? Are these indicators related to other tests such as APTT, PT, and platelet count?”
In the first paragraph of my Discussion section, I provided the analysis the reviewer is seeking. I explicitly stated that this result demonstrates a primarily loss of platelet mediated coagulation kinetics that was likely c-type lectin mediated. This analysis of loss of platelet mediated coagulation would have nothing to do with plasma dependent tests such as the aPTT or PT. Lastly, aggregation caused by a c-type lectin would likely be observed as a decreased platelet count in an EDTA tube. This particular observation is not possible to make unless a viscoelastic instrument (thrombelastograph, ROTEM, etc.) is used. However, to note this within the manuscript in Results would distract from the presentation.
- “While the data in Table 1 is important, it is based on a single animal's results. To maintain the rigor of the paper, it is suggested to remove this from results and briefly describe it in the discussion section with the data placed in supplementary materials.”
I would like to ask for the reviewer’s indulgence in this instance. I have extensively noted the limitations of this one experiment, the results of which should serve as a rare example of finding a spectrum of efficacy for the Ru-based antivenom. To demonstrate that the toxin that affects platelet mediated coagulation is more sensitive to the antivenom than the fibrinogenolytic toxin is unique and intriguing. Rather than an all-or-none response of toxins to antivenom treatment, perhaps other venoms may have a varied response to traditional or Ru-based antivenoms. Thus, I respectfully request to leave this data as it is as an interesting impetus for future research in this matter.
- “The paper involves animal experiments. Has it received the necessary ethical approval? If so, please specify the approval situation or describe the ethical guidelines followed in the study.”
The journal blinds the reviewer to these important details to prevent my identity from being revealed. The manuscript has a statement at the end composed by Toxins: “Institutional Review Board Statement: The authors have got it.” Let me assure the reviewer that my work is of the highest ethical standard with University IACUC oversight.
Reviewer 2 Report
Comments and Suggestions for Authors
The manuscript intends to describe the advantages of the utilization of point-of-care technology to assess the hematological and hemostatic disturbances in snakebite patients in the most reliable manner. The author claims a misalignment between the coagulation status of the snakebite patients and the commonly used laboratory tests, based on the nature of the toxins found in snake venoms, along with the processing time of the laboratory tests, which usually is delayed and does not offer a “realistic” scenario of the envenomed patient. To support this statement, the author also presents unpublished data of the benefits of employing POC technology to understand the hemostatic disturbances caused by snake venoms by using an experimental model of coagulopathy previously published to assess a poorly studied rattlesnake venom.
The manuscript needs to be improved since it is not providing sufficient data to support the idea being reviewed regarding the usage of the POC technology to get an accurate diagnosis of the hemostatic pathophysiology of the snakebite. In this sense, the author needs to explain in depth and provide good evidence supporting how the routinely used anticoagulants (which is the gold standard worldwide to accurately determine hemostatic disturbances in a clinical setting), and the time employed to conduct the classical coagulation assays can modulate or not the different toxins activities based on whether they require or not a co-factor to display their toxic effect, thus altering the final diagnosis about the patient’s conditions. How can this alleged mal alignment affect the final diagnosis and the corresponding patient management and treatment? Could it cause an increased mortality/morbidity rate?
Previous reports have compared the classical coagulation tests with POC technology, showing that the data correlates to each other, however, the findings obtained through POC technology more sensitive. The author should clearly provide more insights into this aspect.
The classification of calcium-dependent and calcium-independent toxins can be too vague for readers with low expertise in the field.
It is well known that North American rattlesnakes present a high diversity in venom composition, which could hint at the different clinical manifestations seen in patients, however the author should explain in better detail how these differences can promote different hemostatic disorders, and why this is relevant for patient management.
Author Response
“The manuscript intends to describe the advantages of the utilization of point-of-care technology to assess the hematological and hemostatic disturbances in snakebite patients in the most reliable manner. The author claims a misalignment between the coagulation status of the snakebite patients and the commonly used laboratory tests, based on the nature of the toxins found in snake venoms, along with the processing time of the laboratory tests, which usually is delayed and does not offer a “realistic” scenario of the envenomed patient. To support this statement, the author also presents unpublished data of the benefits of employing POC technology to understand the hemostatic disturbances caused by snake venoms by using an experimental model of coagulopathy previously published to assess a poorly studied rattlesnake venom.”
“The manuscript needs to be improved since it is not providing sufficient data to support the idea being reviewed regarding the usage of the POC technology to get an accurate diagnosis of the hemostatic pathophysiology of the snakebite.”
I respectfully disagree with the reviewer and ask that he/she reexamine my work. The data concerning calcium-dependent and calcium-independent enzymes/proteins degrading or polymerizing fibrinogen is widely accepted in the basic science space. In fact, I modify my assay system or monitoring system based on the basic biochemistry of these toxins. For example, I designed in vitro assays wherein either time of incubation in citrated plasma or venom concentration is modified to obtain a consistent hematological derangement – typically secondary to effects on fibrinogen as a substrate. My testing paradigm to assess venom toxin activity that is calcium-independent counts on there being an ongoing destruction of fibrinogen before calcium is added to the sample to commence coagulation. I have published dozens of articles concerning these matters. I tried in my introduction to use these basic science facts to explain the disconnect between lab results and patient conditions. I have had several of my poison control colleagues point out that the patients seem fine but the lab values indicate otherwise, which is a real problem when dosing antivenom and worrying about whether they are overdoing it. POC would solve this problem as there would be no ongoing lesion to citrated plasma as I outline in the beginning of my manuscript. This work is a call-to-arms to investigate and likely change clinical practice.
“In this sense, the author needs to explain in depth and provide good evidence supporting how the routinely used anticoagulants (which is the gold standard worldwide to accurately determine hemostatic disturbances in a clinical setting), and the time employed to conduct the classical coagulation assays can modulate or not the different toxins activities based on whether they require or not a co-factor to display their toxic effect, thus altering the final diagnosis about the patient’s conditions.”
I again encourage the reviewer to reevaluate my text. I have been in the hematological space for almost a quarter century and in the thrombelastography/envenomation space for 9 years. I understand the use of citrated or EDTA treated samples are considered the “gold standard” – but for what exactly? These anticoagulants were designed to prevent human proteins and cells from forming clots before arrival to the laboratory. Preventing the conversion of prothrombin to thrombin, a calcium-dependent event, is the way citrate is useful. However, the activation of contact proteins, beginning with FXII, are calcium-independent, so the precursors of prothrombin activation increase over time in citrated samples. I have added some text in the section concerning these issues to emphasize the calcium-dependence of the conversion of prothrombin to thrombin. This is overcome by using very strong activators in the aPTT and PT tests to overshadow any accumulation of activated contact proteins. The same problem occurs with platelet aggregation in EDTA samples that prevent human thrombin from forming but cannot prevent calcium-independent snake venom proteins (e.g., c-type lectins) from causing aggregation that results in a falsely small platelet count. In my experience, the snake venom toxins do not seem to need any special co-factors to exert their effects and damage coagulation. In summary, the faster one can assess blood from an envenomed patient, the more likely the laboratory result will correlate with the clinical condition.
“How can this alleged mal alignment affect the final diagnosis and the corresponding patient management and treatment? Could it cause an increased mortality/morbidity rate?”
I respectfully request that the reviewer reexamine my text. I specifically outlined in the section of the review entitled “Swollen patients, bleeding test tubes” how changes in laboratory values will change as antivenom is administered. I appreciate the comment about patient management and have added a statement to this section: “This malalignment could lead to potential excessive antivenom administration or even unneeded transfusion of platelets/fresh frozen plasma/cryoprecipitate to treat a clinical coagulopathy that does not exist.” Lastly, with regard to mortality/morbidity, rattlesnake envenomation is treated primarily to reduce tissue injury/edema at the bite site and proximal to it in the effected limb. I do not think that the malalignment would increase mortality per se, unless there was a transfusion reaction from a blood product or an anaphylactic reaction if too much antivenom or different antivenom was administered to treat coagulopathy.
“Previous reports have compared the classical coagulation tests with POC technology, showing that the data correlates to each other, however, the findings obtained through POC technology more sensitive. The author should clearly provide more insights into this aspect.”
I appreciate the reviewer’s comment, but after being in the hematological and clinical chemistry space for decades, I have not heard of POC technology being more sensitive. I don’t know what “sensitivity” means as stated by the reviewer. I cannot provide more insight into an issue I know nothing about and would need further insight from the reviewer. As a rule, POC and standard tests only vary by the ability to activate blood coagulation/count platelets, etc., by location and speed of data generation. The activators, data, etc. should be essentially identical.
“The classification of calcium-dependent and calcium-independent toxins can be too vague for readers with low expertise in the field.”
I would like to accommodate the reviewer’s concerns, but the terms calcium-dependent and calcium-independent are anything but vague and are widely accepted in the biochemical literature at large, not just in the toxins space. How can I be more specific concerning the one element, calcium, and its relationship to a chemical reaction, dependent/independent? I cannot make this any less specific, and it seems anything but vague. I am at a loss to substitute some other phrase and thus need to keep my text as it is.
“It is well known that North American rattlesnakes present a high diversity in venom composition, which could hint at the different clinical manifestations seen in patients, however the author should explain in better detail how these differences can promote different hemostatic disorders, and why this is relevant for patient management.”
I have already made this comparison when I contrasted the effects of Western diamondback rattlesnake venom to that of the Eastern diamondback rattlesnake in the section of the manuscript entitled: “A rattle is about the only feature North American rattlesnakes share.” I site reports of human [4-7,12-14] and rabbit [34,35,41] envenomation by both species and explain how the same hematological/thrombelastographic result can be observed after envenomation for entirely different biochemical reasons. I have added the following statement to the Discussion within the concluding paragraph: “The thrombelastographic “finger print” of envenomation will depend on the effects of species-specific toxins, with different combinations of toxin potentially having the same “finger print.” Thus, as with all laboratory assessments, the clinical correlation with likely species responsible for envenomation must always be considered.”
Round 2
Reviewer 1 Report
Comments and Suggestions for Authors
The authors have answered most of the questions. I have no more comments.
Author Response
I see no other comments to address. I thank the reviewer for their suggestions to improve the manuscript.